

# Artificial substrata increase pond farming density of grass carp (*Ctenopharyngodon idella*) by increasing the bacteria that participate in nitrogen and phosphorus cycles in pond water

Zhifei Li, Guangjun Wang, Ermeng Yu, Kai Zhang, Deguang Yu, Wangbao Gong and Jun Xie

Key Laboratory of Tropical and Subtropical Fishery Resource Application and Cultivation, Pearl River Fisheries Research Institute, Chinese Academy of Fishery Sciences, Guangzhou, China
Guangdong Ecological Remediation of Aquaculture Pollution Research Center, Guangzhou, China

Corresponding author
Jun Xie, xiejunhy01@126.com

## ABSTRACT

Aquaculture has become a primary method to produce various aquatic products, and intensive aquaculture technologies have become commercially important. To improve the efficiency of intensive aquaculture per unit area without reducing the growth rate of cultured fish, the present study explored the potential of artificial substrata in ponds. Our results showed that the concentrations of total nitrogen (TN) and total phosphorous (TP) in the ponds with different stocking densities of grass carp were lower than those in the control group in most cases. Further, the feed conversion rate of grass carp was significantly reduced by introducing these artificial substrata, and the culture density could be significantly increased without reducing the growth rates of these fish. Artificial substrata also significantly enriched specific bacteria and changed the structure of the microbiota in pond water. The relative abundance of Proteobacteria was significantly increased, and bacteria closely related to N and P cycles, such as *Hyphomicrobium*, *Chitinimonas*, *Legionella*, *Shewanella*, *Roseiflexus*, and *Planktothrix* were significantly enhanced. These results showed that the artificial substratum could increase TN and TP removal in aquaculture pond water by enriching N and P cycle-related bacteria, thus significantly increasing the specific growth rate of grass carp and significantly reducing their feed conversion rate. Finally, the stocking density of grass carp and the yield per unit area of pond could be increased without reducing the growth rate.

# INTRODUTION

Aquaculture has become a primary method to produce aquatic products, and plays an important role in solving world food shortages and improving the physical qualities of humans (*Bardach, 1985*; *Perschbacher, 2015*). However, with continuous advancements in

global industrialization, aquaculture production space is constantly decreasing. Traditional aquaculture has been unable to meet the rapid growth of human demand for related food. Therefore, improving the yield per unit area could become an effective means to ensure adequate aquaculture production (*Cressey, 2009*). Increasing the farming density is the most commonly used method to increase the yield per unit area, but this ultimately increases the self-purifying load of farming water. Approximately 75% of the feed nitrogen and phosphorus are not utilized and remain as waste in the water (*Gutierrez-Wing & Malone, 2006*), which seriously exceeds the limits of self-purification, thus affecting water quality and indirectly delaying the growth of fish (*Bagley, Bentley & Gall, 1994*; *North et al., 2006*; *Hosfeld et al., 2009*).

Introducing artificial substrata that improve microbial attachment can significantly promote the growth of aquaculture species while restoring water properties *in situ* (*Bo et al., 2010*; *Audelo-Naranjo, Martínez-Córdova & Voltolina, 2010*; *Schveitzer et al., 2013*; *Kumar et al., 2015*; *Li et al., 2017*). The main ways that artificial substrata promote fish growth include the following: reducing ammonia and nitrite concentrations in farming water through the assimilation and dissimilation of microorganisms that adhere to the surface of artificial substrata (*Arndt et al., 2002*; *Zhang et al., 2019*); providing natural feed to farming organisms through the presence of adherent microorganisms on the surface of artificial substrata, such as bacteria, fungi, algae, protozoa, and zooplankton (*Azim et al., 2002*); reducing energy consumption caused by stress reactions through the physical shelter provided by these structures (*Huang et al., 2013*; *Pandey, Bharti & Kumar, 2014*). *Bratvold & Browdy (2001)*, and *Kumar et al. (2017)* both reported that introducing artificial substrata can increase the growth and viability of *Litopenaeus vannamei* and reduce ammonia and nitrite concentrations in farming water.

As one of the most important native Chinese freshwater fish, grass carp (*Ctenopharyngodon idella*) is already the largest freshwater aquaculture product worldwide (*Ni et al., 2014*). The output of grass carp in China alone in 2018 reached 5.345 million tons, accounting for 18.4% of the total output of freshwater aquaculture species (*Administrative Administration of Fisheries and Fisheries of the Ministry of Agriculture and Countryside of China, 2019*), and this provided sufficient amounts of high-quality, low-cost protein for global consumption. To further increase the yield of grass carp per unit pond area without affecting the growth rate and to analyse the potential role of microbial flora that are enriched by the artificial substratum, the present study used grass carp farming density as a variable to determine whether this artificial substratum could effectively alleviate the deterioration in aquaculture water quality caused by the increased culture density.

## MATERIALS AND METHODS

### Experimental design

The experiment was conducted in the precise aquaculture base of the Pearl River Fisheries Research Institute. The experimental period was 60 days. Thirty square cement ponds with of $2 \times 2$ m were employed. An explosion disc was placed in the centre of each pond by continuous aeration 24 h every day, with an approximate airflow of $0.42 \times 10^{-4}$ m$^3$/s. Before

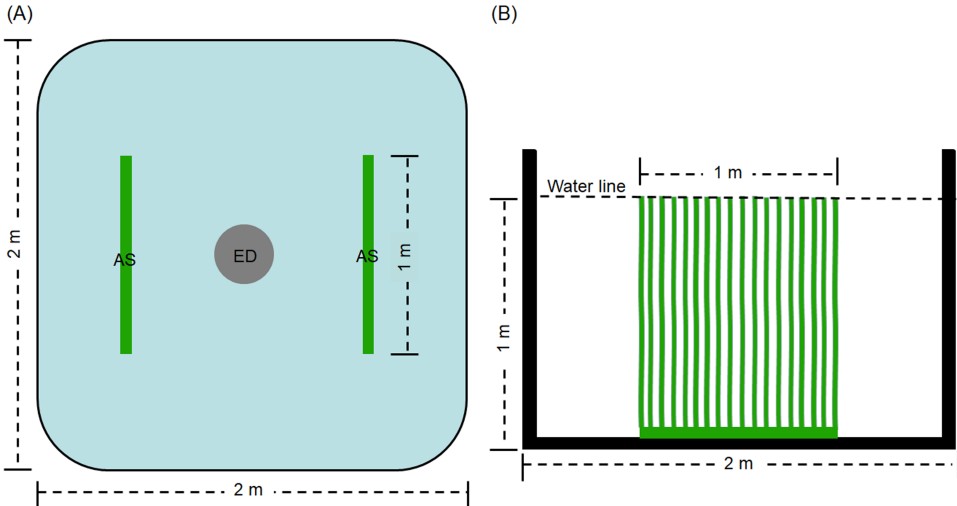

**Figure 1** **Top view (A) and side view (B) of artificial substratum placement in pond.** AS, artificial substratum; ED, explosion disc. Source credit: Zhifei Li and Yankun Cao.

the experiment, the ponds were sterilized for 2 h using 0.1% of potassium permanganate. Then, water was added the ponds to a depth of 1 m. The ponds for treatment groups (D0+S2, D20+S2, D30+S2, D40+S2, and D50+S2, based on density, described as follows) contained two artificial substrata (AquaMats). The polypropylene nonwoven that does not easily decompose and is environmentally friendly was used for the artificial substratum. The size of the artificial substratum was $1 \times 1$ m and its density was described by *Azim et al. (2004)* and *Zhang et al. (2019)*. The artificial substrata were placed as shown in Fig. 1. The ponds for control groups (D0, D20, D30, D40, and D50) did not contain the artificial substratum. There were three replicates for each group and five densities were tested, including 0, 20, 30, 40, and 50 individuals per pond, for both the control and treatment groups. The initial average body weight of the grass carp was $0.35 \pm 0.07$ kg. Commodity feed containing 30% crude protein was administered at 9:00 and 16:00 every day, and the daily feed amount was 3% of the total weight of the grass carp in pond. During the experiment, all ponds were aerated by disc microporous aeration. The pond water was changed according to the pH value of the water. Specifically, the pH value was determined once per day and when this value was less than 6.0, 10% of the pond water was exchanged with aerated tap water.

## Determination of physical and chemical indexes

Starting from the stocking of grass carp, the dissolved oxygen, pH, and temperature in the water were measured daily using the ProPlus portable multiparameter water quality measurer (YSI, USA). Total nitrogen (TN), total phosphorus (TP), ammonia nitrogen ($NH_4^+$-N), nitrate nitrogen ($NO_3^-$-N), nitrite nitrogen ($NO_2^-$-N), and chemical oxygen demand (COD) were measured at 10:00 once every 7 days. Among them, TN and TP contents were measured using the Kjeldahl method and molybdenum blue colorimetry method, respectively (*Lu, 2000*). $NH_4^+$-N, $NO_3^-$-N, and $NO_2^-$-N were measured using

a Pharo100 multi-parameter water quality analyzer (Merck, Germany), and COD was measured with a COD measurer (HACH, USA).

The total exchange ratio of water (TERW) was calculated as follows:

TERW = TVEW/VPWS,

where TVRW is the total volume of exchanged water and VPWS is the volume of pond water at the start.

## Growth performance evaluation

At the end of the experiment, after 24 h of starvation, the pond water was drained and the body weights of grass carp were determined. The specific growth rate (SGR, %/d) and feed conversion rate (FCR, %) were calculated as follows (*Cui & Wootton, 1988*): SGR = {[ln(final weight) − ln(initial weight)] ×100}/experimental days. FCR = feed consumption (dry weight)/(final fish weight (wet weight) − initial fish weight (wet weight)).

## Microbiota collection, DNA extraction, and sequencing

Each 200-ml pond water sample was collected at the end of the experiment and filtered using a GF/C filter with a 0.22-μm pore size. The filter was cut into fragments and placed in a 50-ml sterile centrifuge tube for DNA extraction. Three grams of each artificial substratum sample were also weighed and added to 200 ml of sterile water. The mixture was then vortexed for 2 h. The mixture was centrifuged at 2,500 × g for 15 min at 4 °C and the suspension was collected for DNA extraction. For this, the suspensions were filtered using GF/C filters with 0.22-μm pore size. The filter was cut into fragments and placed in a 50-ml sterile centrifuge tube for DNA extraction. Bacterial DNA was extracted using a kit for the extraction of bacterial DNA from water (Omega, Norcross, GA, USA). DNA concentration and purity were evaluated with 1% agarose gels. Based on the concentration, DNA was diluted to 1 ng/μl with sterile water for further amplification. The V4 hypervariable region of the 16S rRNA gene was amplified using the 515F and 806R primers with sample-specific barcodes (*Yan et al., 2016*). PCR was performed in 30-μl reaction volumes with 15 μl of Phusion High-Fidelity PCR Master Mix (New England Biolabs, Ipswich, MA, USA), each primer at 0.2 μM, and approximately 10 ng of template DNA. Thermal cycling conditions were as reported previously (*Li et al., 2017*). The PCR products were mixed at ratios of equal density and the mixed PCR products were purified using the Gene JET Gel Extraction Kit (Thermo Scientific, Waltham, MA, USA). Sequencing libraries were constructed using the NEB Next Ultra DNA Library Prep Kit for Illumina (New England Biolabs), according to the manufacturer's recommendations, and index codes were added. Finally, the libraries were sequenced using the Illumina MiSeq platform and 250-bp paired-end reads were generated. MiSeq sequencing was conducted by Novogene Co. (Beijing, China).

The paired-end reads from the raw DNA fragments were merged using FLASH software. The merged tags were assigned to each sample according to the sample-specific barcodes, and the low-quality sequences were detected and removed using QIIME 1.9.0 software according to previous reports (*Li et al., 2017*; *Huang et al., 2018*; *Ni et al., 2017*; *Ni et al., 2019*). Chimeric sequences were detected and removed using Uchime algorithm (*Edgar et*

*al., 2011*) embedded to QIIME 1.9.0 software. QIIME 1.9.0 software (*Caporaso et al., 2010*) and the vegan package (*Dixon, 2003*) in the R platform (*R Core Team, 2014*) were used to analyze alpha (within samples) and beta (among samples) diversity. Sequences with ≥97% similarity were assigned to the same operational taxonomic unit (OTU) using UPARSE software (*Edgar, 2013*). Representative sequences for each OTU were selected and the RDP classifier was used to annotate them with appropriate taxonomic information (*Wang et al., 2007*). In-house Perl scripts were used to analyse alpha and beta diversity.

All merged DNA sequences have been submitted to the genome sequence archive database (https://bigd.big.ac.cn/) under the accession number CRA001827 (https://bigd.big.ac.cn/gsa/s/Z6ZLD9E9).

## Data analysis

Data were recorded and pre-treated using Excel. Data are shown as the mean ± standard error. Principle component analysis (PCoA) was conducted using QIIME 1.7.0, non-parametric multivariate analysis of variance (PERMANOVA; *Anderson, 2001*) was conducted using R software with the vegan package (*Dixon, 2003*), and one-way ANOVA and t-tests were conducted using R software with the base packages. Linear discriminant analysis effect size (LEfSe) was conducted using the Galaxy platform as our previous description (*Li et al., 2019*). $P$ values <0.05 were considered significant.

# RESULTS

## Artificial substrata significantly increase the pond farming density of grass carp

The water temperatures of ponds were not significantly different during the experiment (one-way ANOVA, $F = 0.736$, $p = 0.676$). Dissolved oxygen in the pond water was closely and negatively correlated with farming density. The pH values of ponds without farmed fish ranged from 7.9 to 8.4, which were significantly higher than those of ponds housing farmed fish (from 5.5 to 7.5; one-way ANOVA, $p < 0.05$; Table S1 ). No fish died during the experimental process. Except for those in the pond without grass carp (D0 and D0S), the concentrations of TN and TP in other ponds fluctuated and increased with culture time from the beginning of the experiment, but remained in specific range after the middle of the experiment (Figs. S1 and S2). The increase in TN concentration was mainly due to the significant increase in nitrate concentration in the pond water (Fig. S3), because nitrite and ammonia nitrogen concentrations did not gradually increase during the experimental process, but rather fluctuated within a certain concentration range (Figs. S4 and S5). In most cases, TN and TP in the ponds that contained artificial substrata were lower than those in the control ponds at the later stage of the experiment (Figs. S1 and S2). Similar results were also obtained regarding the concentrations of nitrate, nitrite, and ammonia nitrogen, especially for the high stocking density ponds (D40 and D50; Figs. S3–S5).

With an increase in grass carp farming density, the TERW and FCR of both treatment and control groups increased, whereas the SGR of grass carp decreased gradually (Fig. 2). The SGR of grass carp was significantly increased with the presence of artificial substrata (Paired $t$-test, $t = -3.070$, $p = 0.011$), and the FCR was significantly reduced (Paired

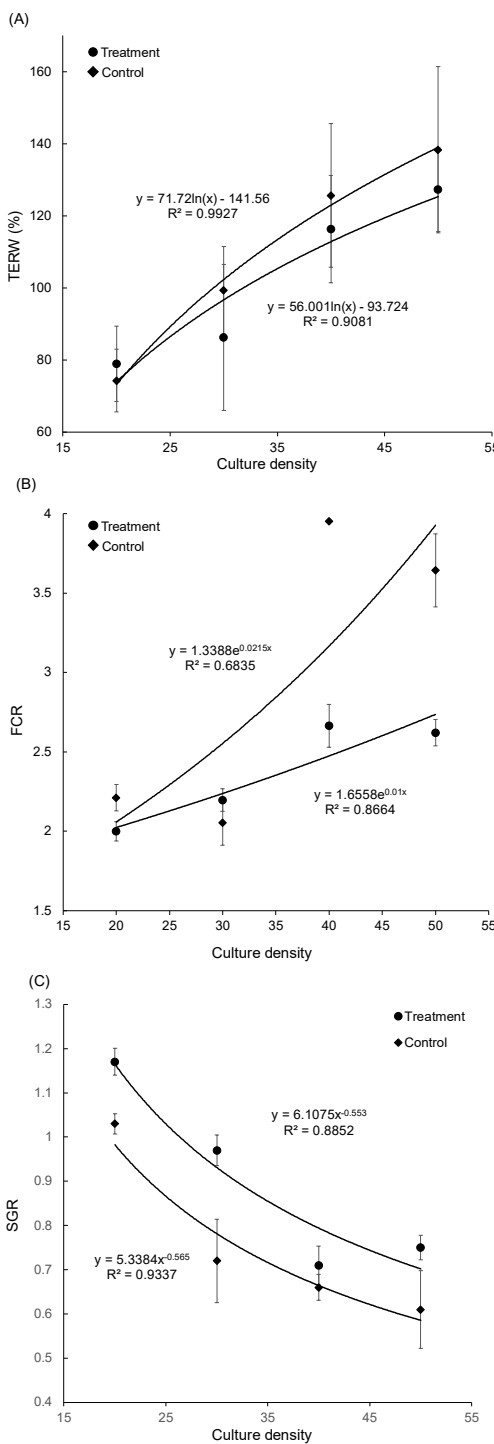

**Figure 2  Correlation between culture density of grass carp and total exchange ratio of water (A), feed conversion rate (B), and specific growth rate (C).** TERW, total exchanged ratio of water; FCR, feed conversion rate; SGR, specific growth rate.

$t$-test, $t = 3.045$, $p = 0.011$), which indicated that the introduction of artificial substrata into aquaculture ponds promotes the growth of grass carp. According to the fitting formulas $SGR_{treatment} = 6.1075$ (farming density)$_{treatment}^{-0.553}$ and $SGR_{control} = 5.3384$ (farming density)$_{control}^{-0.565}$, the artificial substrata were able to increase stocking density to 10.68 grass carp per pond while maintaining an SGR of 0.8. Thus, the estimated stocking density in the pond could be increased by 0.88 kg/m$^3$ (2.67 individuals/m$^3$).

## Artificial substrata significantly increase the bacteria that participate in nitrogen and phosphorus cycles

To determine whether the introduction of artificial substrata could increase the microbial species involved in nitrogen metabolism in pond water, the microbiota community structures of 30 pond water samples (three duplicate samples in five treatment groups and five control groups) and 15 artificial substrata were analysed. In total, 2,557,442 (56,832.04 ± 1,202.30) high-quality sequences were obtained. To exclude the influence of sequencing depth, 28,604 sequences were randomly resampled from each sample for further analysis. A total of 6782 OTUs were obtained based on 97% sequence similarity. These sequences belonged to 51 phyla, except for a few (1.03 ± 0.06%) sequences that could not be classified at the phylum level. Acidobacteria, Actinobacteria, Armatimonadetes, Bacteroidetes, Chlorobi, Chloroflexi, Cyanobacteria, Firmicutes, Fusobacteria, Gemmatimonadetes, Nitrospirae, Planctomycetes, Proteobacteria, Synergistetes, and Verrucomicrobia were the dominant phyla, for which relative abundance was more than 1% in at least one sample (Fig. 3A). The relative abundances of Actinobacteria, Chlamydiae, Fusobacteria, and Planctomycetes in pond water were significantly decreased with the treatment, whereas the relative abundance of Proteobacteria was significantly increased (Fig. 3B). This was closely correlated with the significant enrichment of Proteobacteria induced by the artificial substrata (Fig. 3B).

Except for a few sequences (1.03 ± 0.06%), 928 prokaryotic genera were obtained, of which 320 were dominant genera (their relative abundances were more than 0.1% in at least one sample). PCoA results showed that the introduction of an artificial substratum could not only result in the formation of microbial communities in the artificial substrata that were distinct from those in pond water, but could also significantly change the composition of microbiota in the pond water (PERMANOVA, $F = 6.77$, $p = 0.005$). Interestingly, the composition of microbiota in pond water with artificial substrata was more similar to that of the artificial substrata (Fig. 4).

LEfSe based on dominant genera showed that those that were significantly enhanced by the artificial substratum gradually decreased with increasing grass carp stocking densities, mainly because the microbiota on the artificial substrata were closer to those of the pond water (Fig. 4). Significantly enriched bacteria on the artificial substrata mainly comprised Proteobacteria, in which *Ideonella*, *Nordella*, *Hyphomicrobium*, *Pseudoduganella*, and *Chitinimonas* were the most significantly enriched (they were significantly enriched with at least two farming densities; Fig. 5). In addition, *Perlucidibace*, *Polynucleobacter*, *Legionella*, *Romboutsia*, *Shewanella*, *Roseiflexus*, *Planktothrix*, and *Limnothrix* in the pond water with artificial substrata were significantly enhanced (Fig. 5).

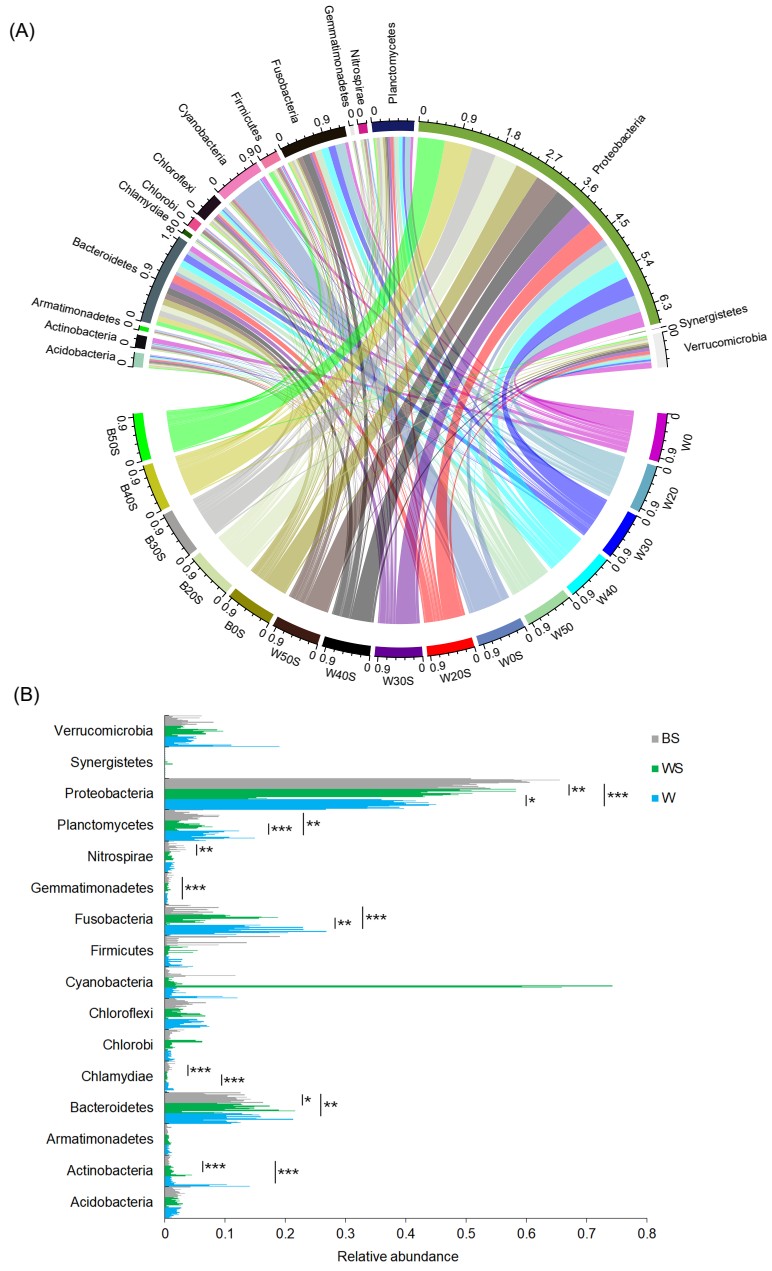

**Figure 3 Changes in the dominant phyla between the ponds containing artificial substrata and controls.** A circular layout (A) and bar diagram (B) are shown. BS, microbiota from artificial substrata; WS, microbiota from the pond water with artificial substrata; W, microbiota from the control pond water.

## DISCUSSION

High-density culture in ponds not only reduces the growth rate of cultured fish, but also increases their FCR, and increases the accumulation of TN and TP in pond water and sediment (*Rahman et al., 2005*; *Besson et al., 2016*). The accumulation of TN and TP, and especially the increase in nitrite and ammonia nitrogen, will lead to the slow growth of

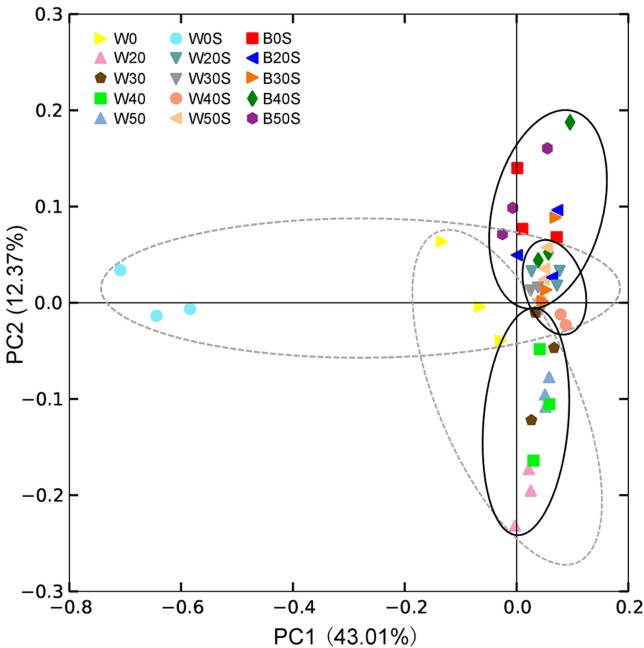

**Figure 4 Principle component analysis (PCoA) profile showing the differences in microbiota among pond water and artificial substratum samples.** The dotted line ellipses depict the microbiota samples from pond water without farmed fish and with artificial substrata, whereas the black solid line ellipses depict the microbiota samples in pond water and artificial substrata, eliminating the samples without farmed fish.

cultured fish (*Ni et al., 2018*). In the present study, our results also showed that the growth rate of grass carp decreased gradually (Fig. 2C) and that the FCR increased continuously (Fig. 2B) with an increase in the culture density of grass carp. We further demonstrated that the presence of an artificial substratum could effectively increase the culture density while maintaining the growth rate of grass carp (Fig. 2C), ultimately increasing the yield per unit pond area. Considerable production gains and reduced FCR were reported when using an artificial substratum, based on the culture of *Farfantepenaeus paulensis* (*Ballester et al., 2007*), *Litopenaeus vannamei* (*Audelo-Naranjo, Martínez-Córdova & Voltolina, 2010*), and *Panaeus monodon* (*Anand et al., 2013*), as this increases the available surface area of biofilm and reduces the negative effects of fish overcrowding. Similarly, our results showed the importance of an artificial substratum in aquaculture ponds. Based on the fitting curve equation obtained in the present study, under the premise of guaranteeing the SGR, the stocking density of grass carp could be increased by including artificial substrata. Taking the SGR of 0.8 as an example, the stocking density could be increased by 0.88 kg/m$^3$ and the stocking biomass could be increased by 37%.

The accumulation of TN and TP in pond water is considered an important reason for slower growth rates in aquaculture (*Rahman et al., 2005*; *Besson et al., 2016*; *Ni et al., 2018*). In the present study, our results showed that TN and TP in pond water fluctuated but increased with culture time (Figs. S1 and S2). The increasing trend for TN was mainly caused by an increase in nitrate-nitrogen (Fig. S3), whereas an increase in ammonia-nitrogen and

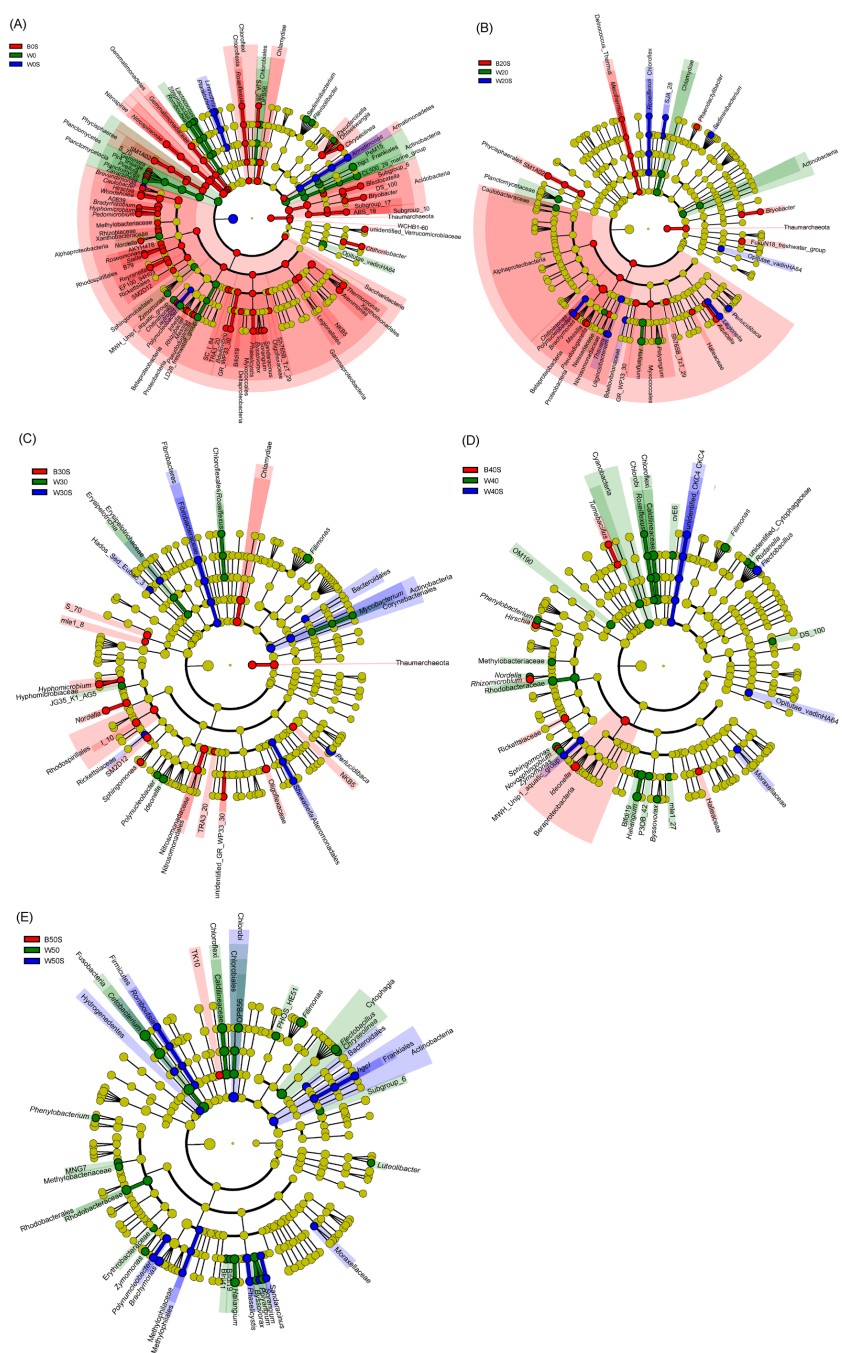

**Figure 5 Linear discriminant analysis effect size (LEfSe) profiles showing the significant differences in microbiota among pond water and artificial substratum samples.** (A–E) Significantly different taxa in the samples from the pond without farmed fish, farmed fish at five individuals/m², farmed fish at 7.5 individuals/m², farmed fish at 10 individuals/m², and farmed fish at 12.5 individuals/m², respectively. Red, green, and blue indicate significantly different taxa in the artificial substratum microbiota, control pond water, and pond water from the ponds containing artificial substrata, respectively. Source credit: Zhifei Li and Yankun Cao.

nitrite was not obvious during the experiment (Figs. S4 and S5). This might be due to the continuous explosion during the experiment, in addition to the fact that high dissolved oxygen promotes the growth of heterotrophic bacteria through nitrification, which results in the oxidation of ammonia-nitrogen and nitrite to form nitrate through nitrification (*Ebeling, Timmons & Bisogni, 2006*; *Kumar et al., 2017*). *Asaduzzaman et al. (2008)* also showed that the total ammonia-nitrogen concentration in the treatment pond can be controlled at a lower level than that in the control pond after the addition of a substrate, implying that the substrate supports the growth of nitrifying bacteria and algae, which in turn reduces total ammonia-nitrogen and $NO_2^-$-N levels. In addition, our results also showed that in most cases, the concentration of TN and TP in pond water could be reduced by the presence of artificial substrata; nitrite, ammonia-nitrogen, and nitrate also showed a similar trend, especially for grass carp cultured at a high culture density (D40 and D50), in accordance with the results of previous studies (*Azim et al., 2002*; *Asaduzzaman et al., 2008*; *Kumar et al., 2015*; *Henares et al., 2015*). These results thus showed that water quality might be very important for the growth of aquaculture species, suggesting that water quality control should be strictly maintained during this process.

Biofilms that effectively promote N and P metabolism, maintaining these elements at low concentrations in freshwater ecosystems, have been identified (*Flemming & Wingender, 2010*; *Li et al., 2017*). Further, artificial substrata have been used in ponds to enhance the formation of biofilms that aid in controlling N and P (*Li et al., 2014*; *Yu et al., 2016*). Our previous study also showed that adding artificial substrata to an aquaculture pond could create a habitat for denitrifiers and phosphorus-removing bacteria (*Li et al., 2017*). In the present study, bacteria involved in N and P cycles such as *Hyphomicrobium* (*Sperl & Hoare, 1971*; *Van der Drift & De Windt, 1983*; *Brooke, Duchars & Attwood, 1987*; *Kloos et al., 1995*), *Chitinimonas* (*Chang et al., 2004*), *Legionella* (*Keen & Hoffman, 1984*), *Shewanella* (*Al-Harbi & Uddin, 2006*; *Hau & Gralnick, 2007*; *Fredrickson et al., 2008*), *Roseiflexus* (*Wawrik et al., 2011*; *Penton et al., 2013*; *Gerbl et al., 2014*; *Hug et al., 2015*; *Wang et al., 2018*), and *Planktothrix* (*Zotina, Köster & Jüttner, 2003*; *Davis et al., 2015*) were enhanced on the artificial substrata and in associated pond water (Fig. 5). These results implied that such artificial substrata can reduce TN and TP levels in pond water by enriching bacteria involved in N and P cycles, thus alleviating the inhibitory effects of high TN and TP on the growth of grass carp.

## CONCLUSION

The SGR of grass carp was significantly increased and the FCR was significantly reduced by introducing artificial substrata. This was thought to be mainly due to changes in microbial community structure in the pond water, which resulted in the enrichment of bacteria involved in N and P cycles, as well as the reduction of N and P concentrations in pond water.

## ACKNOWLEDGEMENTS

The authors thank anonymous technicians at Guangdong Meilikang Bio-Science Ltd. (China) for assistance with data re-analysis and drawing of figures.

### Funding

This work was supported by the National Natural Science Foundation of China (31802348), the Science and Technology Planning Project of Guangdong Province, China (2016A020210023), the Natural Science Foundation of Guangdong Province, China (2015A030313699), and the Modern Agro-industry Technology Research System (grant number CARS-45-21). The funders had no role in study design, data collection and analysis, decision to publish, or preparation of the manuscript.

### Grant Disclosures

The following grant information was disclosed by the authors:
National Natural Science Foundation of China: 31802348.
Science and Technology Planning Project of Guangdong Province, China: 2016A020210023.
Natural Science Foundation of Guangdong Province, China: 2015A030313699.
Modern Agro-industry Technology Research System: CARS-45-21.

### Competing Interests

The authors declare there are no competing interests.

### Author Contributions

- Zhifei Li conceived and designed the experiments, performed the experiments, analyzed the data, prepared figures and/or tables, authored or reviewed drafts of the paper, approved the final draft.
- Guangjun Wang performed the experiments, authored or reviewed drafts of the paper, approved the final draft.
- Ermeng Yu performed the experiments, contributed reagents/materials/analysis tools, prepared figures and/or tables, approved the final draft.
- Kai Zhang analyzed the data, prepared figures and/or tables, approved the final draft.
- Deguang Yu conceived and designed the experiments, prepared figures and/or tables, authored or reviewed drafts of the paper, approved the final draft.
- Wangbao Gong analyzed the data, contributed reagents/materials/analysis tools, prepared figures and/or tables, approved the final draft.
- Jun Xie conceived and designed the experiments, authored or reviewed drafts of the paper, approved the final draft.

### Data Availability

All merged DNA sequences are available at the Genome Sequence Archive database: CRA001827 (https://bigd.big.ac.cn/gsa/s/Z6ZLD9E9.).
## Supplemental Information

Supplemental information for this article can be found online at http://dx.doi.org/10.7717/peerj.7906#supplemental-information.

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
