# Peer review of "Artificial substrata increase pond farming density of grass carp (Ctenopharyngodon idella) by increasing the bacteria that participate in nitrogen and phosphorus cycles in pond water"

_PeerJ, doi:10.7717/peerj.7906_

## Round 0.1 · original submission · Major Revisions

Dear Authors,

The reviewers have commented on your above paper. They indicated that it is not acceptable for publication in its present form.

However, if you feel that you can suitably address the reviewers' comments (included), I invite you to revise and resubmit your manuscript. Special attention is needed on the suggested modifications related to figures and references.

·

Basic reporting

The manuscript is written clearly and unambiguously. Professional English used throughout.

The manuscript includes sufficient introduction and background. Relevant prior literatures are appropriately referenced.

The structure of the manuscript conforms to an acceptable format of 'standard sections'. However, the panels of Figure 2, Figure 3(A) and Figure 5 are too small to read. Please reorganized the panles and appropriately adjust the dpi and size of each panel.

The merged sequence data have been made available.

The manuscript includes all results relevant to the hypothesis.

Experimental design

The original primary research within aims and scope of the journal.

The research question is well difined, relevant and meaningful. And the study contributes to filling that gap.

The investigation was conducted rigorously and to a high technical standard.

The methods should be supplemented. Line 82: "An explosion disc was placed in the centre of each pond." However, the authors did not specify the amount of aeration per day. Line 111-115. Please add references of the calculation of SFR and FCR. Line 141. According to what I know, QIIME1 integrated multiple softwares and algorithms. Uchime algorithm (Edgar et al., 2011, Bioinformatics, 27: 2194-2200) commonly used to identify and remove the chimera sequences in QIIME1. Therefore, please further elaborate which specific software or algorithm was used in the chimeric sequence analysis. Line 147. Please submit the in-house Perl scripts to a public script database or using QIIME reanalyzes the alpha and beta diversity, as QIIME1 also can analyse the alpha and beta diversity.

Validity of the findings

The manuscript reported the potential of artificial substrata to improve the efficiency of intensive aquaculture per unit area and characterized the microbiota compositions on the artificial substrata and in pond water. Considering the important role of grass carp on the aquatic products and the aquaculture production space is constantly decreasing, the manuscript provided a valuable technical method to increase the density and yield of grass carp.

All underlying data have been provided, and they are robust, statistically sound, and controlled.

Conclusions of the manuscript are well stated, linked to original research question.

Additional comments

The manuscript is clearly written in professional, unambiguous language. The organization of the manuscript is logical. I recommend accepting the manuscript for publication. However, it has some minor errors which should be improved upon before publication.
Line 82: "An explosion disc was placed in the centre of each pond." However, the authors did not specify the amount of aeration per day.
Line 111-115. Please add references of the calculation of SFR and FCR.
Line 141. According to what I know, QIIME1 integrated multiple softwares and algorithms. Uchime algorithm (Edgar et al., 2011, Bioinformatics, 27: 2194-2200) commonly used to identify and remove the chimera sequences in QIIME1. Therefore, please further elaborate which specific software or algorithm was used in the chimeric sequence analysis.
Line 147. Please submit the in-house Perl scripts to a public script database or using QIIME reanalyzes the alpha and beta diversity, as QIIME1 also can analyse the alpha and beta diversity.
Reference list: Please make sure the format of the references is correct. Beacuse I found the formats of reference "Administrative Administration of Fisheries and Fisheries of the Ministry ..." and those of "Fisheries Bureau of Agriculture Ministry of China. ..." seem different.
Figure 2: The panels are too small to read. I suggest longitudinally arrange the panels and appropriately adjust the dpi and size of each panel.
Figure 3(A): The panel is too small to read. I suggest appropriately adjust the dpi and size of the panel, or longitudinally arrange the panels of Figure 3.
Figure 5: The panels are too small to read. Please reorganized the panles and appropriately adjust the dpi and size of each panel.

Reviewer 2 ·

Basic reporting

no comment

Experimental design

No information is provided about the chemical nature of the artificial substrata. Without this information will makes it cannot be replicated by others. So it is especially important to add this necessary information in the Materials and methods section.

Validity of the findings

no comment

Additional comments

This manuscript with clear presentation of experimental results. The authors provided adequate background of the study with references that are relevant to artificial substratum, and the hypothesis and objectives of the study are clear.

“artificial substratum” and “artificial substrata” should be kept consistancy.

Please make sure that all the LEfSe profifiles and figure 2 described in the text are visible in the figures. I suggest including high magnification inset or another figure.

---

## Round 0.2 · accepted · Accept

Thank you very much for improving your paper.

Reviewer 2 ·

Basic reporting

The English is clear and all the references are necessary.

All data including figures and tables are also clear.

Experimental design

This study within Aims and Scope of the PeerJ and the research question is interesting. Methods are described sufficient now.

Validity of the findings

Findings of this study is important for aquaculture and environment.

Additional comments

All my original comments and suggestion were fully addressed in the revision, I have no additional comment.